# Clonal heterogeneity and antigenic stimulation shape persistence of the latent reservoir of HIV

**Marco Garcia Noceda**[1]ꝏ, **Gargi Kher** [2]ꝏ, **Shikhar Uttam**[2,3], **John P. Barton** [1,2,4]*

1 Department of Physics and Astronomy, University of California, Riverside, California, United States of America, 2 Department of Computational and Systems Biology, University of Pittsburgh School of Medicine, Pittsburgh, Pennsylvania, United States of America, 3 UPMC Hillman Cancer Center, Pittsburgh, Pennsylvania, United States of America, 4 Department of Physics and Astronomy, University of Pittsburgh, Pittsburgh, Pennsylvania, United States of America

ꝏ These authors contributed equally to this work.

* jpbarton@pitt.edu

**Data availability statement:** All data and code used in this study are freely available at the

## Abstract

Drug treatment can control HIV-1 replication, but it cannot cure infection. This is because of a long-lived population of quiescent infected cells, known as the latent reservoir (LR), that can restart active replication even after decades of successful drug treatment. Many cells in the LR belong to highly expanded clones, but the processes underlying the clonal structure of the LR are unclear. Understanding the dynamics of the LR and the keys to its persistence is critical for developing an HIV-1 cure. Here we develop a quantitative model of LR dynamics that fits available patient data over time scales spanning from days to decades. We show that the interplay between antigenic stimulation and clonal hetero-geneity shapes the dynamics of the LR. In particular, we find that large clones play a central role in long-term persistence, even though they rarely reactivate. Our results could inform the development of HIV-1 cure strategies.

## Author summary

When people with HIV take effective treatment, the virus can no longer be detected in the blood, but a small number of infected cells persist in a dormant state, forming what is known as the latent reservoir. This hidden pool of virus is the main barrier to a cure because it can reactivate if treatment is stopped. In this study, we developed a mathematical model that simulates the behavior of individual infected cell lineages, or "clones," to understand how the reservoir changes over time. Our model incorporates two key forces: the chance that a latent virus reactivates and the signals from the immune environment that drive infected cells to divide. Our results show that the reservoir becomes dominated by a few large, long-lived clones that rarely reactivate, while smaller, more dynamic clones are gradually lost. Importantly, we find that low-level viral replication during

GitHub repository https://github.com/bartonlab/paper-HIV-latent-reservoir.

**Funding:** The author(s) received no specific funding for this work.

**Competing interests:** The authors have declared that no competing interests exist.

treatment does not necessarily lead to viral evolution — a distinction that challenges common assumptions and helps explain why the virus remains genetically stable in individuals on long-term treatment. Our findings suggest that eliminating the reservoir may require strategies that go beyond reactivation, targeting the large and persistent clones that silently maintain the infection.

## Introduction

Human immunodeficiency virus (HIV-1) actively replicates in CD4$^+$ T cells [1,2]. During the infection process, viral RNA is reverse-transcribed into DNA, which is then incorporated into the genome of the host cell [3]. This integrated viral DNA is referred to as a provirus. Most infections result in the rapid production of new viruses, leading to the death of these infected cells within a few days [4]. However, in a fraction of cells, HIV-1 is capable of lying in a dormant, "latent" state [5]. While antiretroviral therapy suppresses active HIV-1 replication, it is unable to eliminate latently infected cells or their integrated proviruses [6]. And while many latent proviruses are defective, some remain capable of reactivation, resulting in a quick return to active infection if antiretroviral treatment (ART) is interrupted, even in individuals who have undergone effective drug treatment for many years [7,8]. This population of long-lived, latently infected cells, known as the latent reservoir (LR), therefore presents the major barrier to an HIV-1 cure.

Understanding factors that contribute to LR persistence could greatly contribute to HIV-1 cure efforts. However, it is difficult to obtain a comprehensive picture of LR dynamics from direct measurements due to its small size. For typical HIV-1-infected individuals, roughly one in $10^4$ CD4$^+$ T cells are latently infected, and active HIV-1 replication occurs in only around 1% of these latently infected cells in viral outgrowth experiments [9–11]. Thus, latently infected cells, especially ones that can readily reactivate, are rare. Subsequent studies have also found that multiple rounds of stimulation can prompt latent cells that initially remained dormant to reactivate, making it difficult to determine the total number of latently infected cells that are capable of reactivation [11,12].

Despite these challenges, recent work has provided insights into the dynamics and heterogeneity of the LR. Subsets of cells bearing integrated HIV-1 can undergo clonal expansion in patients receiving suppressive ART [13,14]. The degree of expansion of clones as well as their persistence varies greatly and is associated with the specific integration sites [13,14,14] as well as stimulation by antigens [15,16]. Levels of HIV-1 expression in latently infected cells differ, and there appears to be progressive selection for more transcriptionally silent integration sites during long-term ART [17,18]. Persistence can also occur via T cell proliferation, including both homeostatic proliferation [19] and in response to antigen [15,16,20,21]. While many highly-expanded clones contain defective proviruses [22,23], at least half of the cells carrying intact proviruses also belong to expanded clones [23–27]. A strong negative correlation has also been observed between clone size and reactivation rate in viral outgrowth assays [25]. As patients remain on ART for long times, the diversity of observed clones decreases and the proportion of HIV-1 proviruses in the largest clones progressively increases [28,29].

Mathematical modeling has also provided insights into the LR, with some model predictions validated in experiments [30]. Studies investigating the relationship between latently infected cells and plasma viremia during ART [31–36] suggest that as long as ART is marginally effective, the persistence of latent virus is most strongly influenced by the longevity of infected cells and the rate at which they reactivate. Recent modeling work has also

suggested that uneven homeostatic proliferation of latently infected cells early in infection may lead to the observed spread in clone sizes in the LR [37].

To gain a deeper understanding of the persistence of the latent reservoir in HIV-infected individuals, it is important to develop mathematical models that reflect the biological mechanisms that govern its dynamics. The LR is composed of diverse clones with different T cell receptors (TCRs), which affect their activation potential and antigen specificity. Moreover, these clones have distinct viral integration sites, which influence their transcriptional activity and reactivation probability. These factors contribute to the clonal heterogeneity of the LR and its persistence. Thus, there is a need to incorporate more comprehensive and biologically motivated features of clonal heterogeneity, which are not typically incorporated in existing mathematical models of the LR.

We addressed this challenge by developing a novel stochastic model of LR dynamics that explicitly accounts for clonal heterogeneity. We consider genetic changes in HIV-1 sequences and variable probabilities of reactivation, while also incorporating the effects of antigenic stimulation on latently infected clones with different TCRs. The dynamics of these clonal populations are integrated with interactions between free viruses, susceptible cells, and cells that are actively infected. We model multiple latently infected clones, integrated into distinct T cell clones with different TCRs and thus different responses to antigen.

Our model recapitulates experimentally observed features of HIV-1 infection while also providing insights into LR structure, dynamics, and persistence. We recover the decay kinetics of HIV-1 RNA in blood, HIV-1 DNA in peripheral blood mononuclear cells (PBMCs), and latent cells that reactivate upon stimulation, which occur over widely-varying time scales (days, months, and years) following the start of ART [38–40], without the use of time-varying parameters (S1 Table). Among other findings, our model reproduces the observation of defective proviruses in highly expanded clones [22] and the negative correlation found between clone size and reactivation probability for patients who have undergone ART treatment for many years [25]. Stimulation by antigens combined with heterogeneous reactivation rates for different clones leads to a broad distribution of clone sizes, which are stratified by their reactivation rates. Over long times, we find that the LR becomes progressively more concentrated on a small number of clones with low reactivation rates, which play a key role in LR persistence. These insights could inform the development of new therapeutic approaches to reduce the size of the LR and achieve a functional HIV-1 cure.

## Results

**Stochastic model incorporating LR heterogeneity.**   Our model blends elements from multiple prior mathematical studies [30–32,37,38,41–46]. We consider four main populations: uninfected CD4+ T cells (T), productively infected activated CD4+ T cells (A), latently infected resting CD4+ T cells (L), and HIV-1 virions (V) (Fig 1A). All cells have finite lifespans determined by their respective death rates (see Methods for a complete list of parameters and supporting references). We used a constant replacement rate $\lambda_T$ to approximate the replenishment of uninfected target cells from the thymus [45,47,48]. The rate of production of virions is given by the product of the death rate of actively infected cells $\mu_A$ [49] and the viral burst size $n$ [50]. Virions are then cleared at a constant rate $c$ [51].

HIV-1 virions can infect susceptible CD4+ T cells. A small fraction of infections, $p_{\text{def}}$, will result in defective integrated proviruses due to effects like large deletions or hypermutation. During ART, this leads to proviruses with fatal defects outnumbering intact proviruses by a factor of 10–50 to 1 [28]. HIV-1 also mutates during infection due to error-prone reverse transcription, with an estimated error rate of $3 \times 10^{-5}$ per base per replication cycle [52–54].

Given the length of the HIV-1 genome, this implies that approximately a third ($p_{mut}$) of successful infection events will lead to a mutation. Finally, a fraction $p_L$ of infection events will result in latent rather than active infection. We fit $p_L$ such that the HIV-1 DNA per $10^6$ PBMCs at the beginning of ART was in the range $10^3$–$10^4$, consistent with patient data [38]. Consistent with reported HIV-1 RNA levels during ART [39,55] and following previous modeling studies [35], we assume that viral replication is attenuated but not perfectly suppressed during ART.

To account for the heterogeneity of latently infected cells, we modeled each latently infected clone individually, including the expansion and antigen-driven proliferation of individual clones (Fig 1B). Clones are defined as latently infected cells with identical TCRs, integrated proviruses, and integration sites. Each time a new clone is created through a latent infection event, it is assigned a random probability of reactivation (Methods), following the observation that integration is stochastic and different integration sites can affect the capacity for reactivation [56]. In addition, each clone is stimulated by a background concentration of antigen that fluctuates in time (Methods), inspired by past models of T cell repertoire dynamics [46]. We use the same homeostatic death and proliferation rates for all clones; however, stochastic differences in antigenic stimulation drive differences in clonal proliferation. As a simplifying assumption, we model all susceptible T cells as identical, and we only consider antigen-driven proliferation dynamics for latently infected cells. Our assumption that T cell activation and recognition of antigens can drive HIV-1 reactivation, and that reactivation is

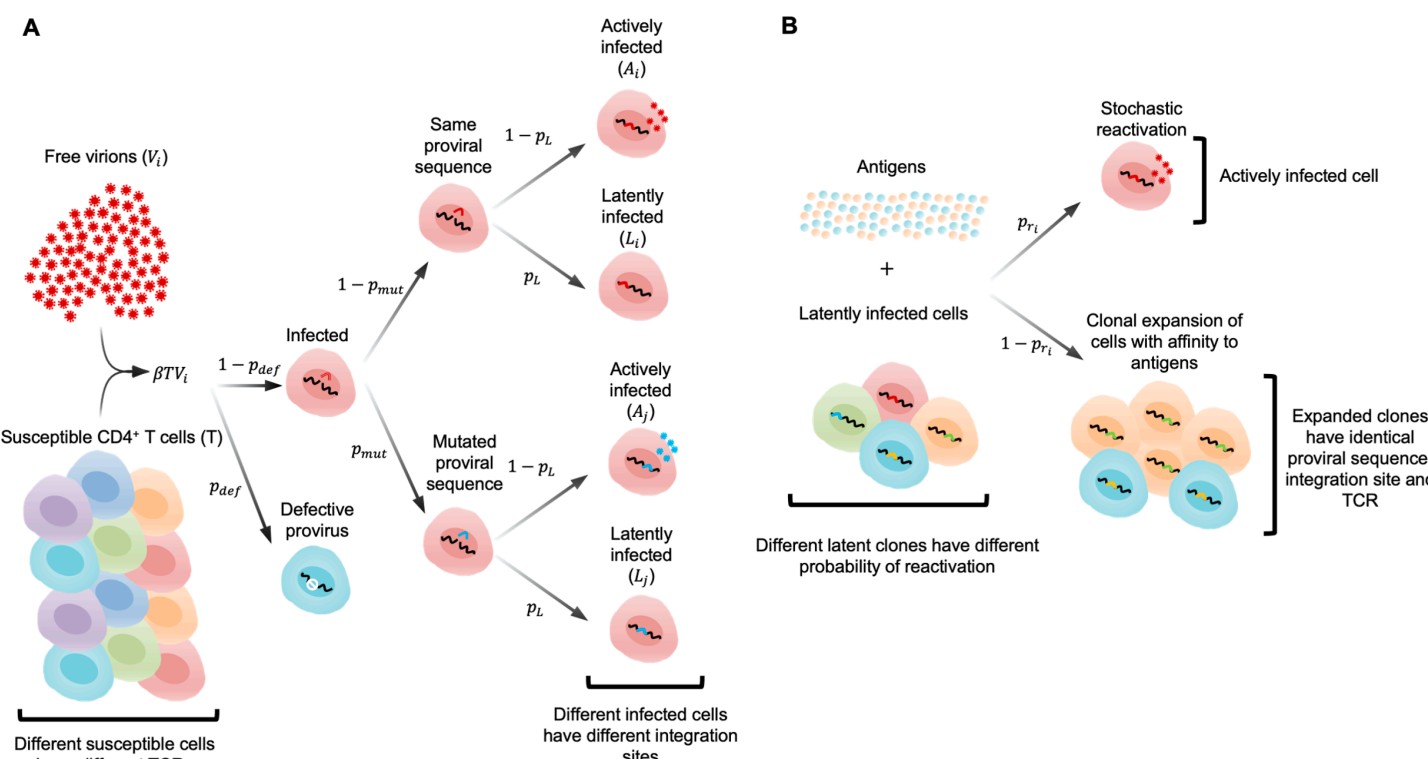

**Fig 1. Model schematic.** (**A**) When a new cell is infected, it forms a new "clone" with a specific TCR sequence. The generation of defective proviruses, point mutations, and active or latent infection are all determined by chance. (**B**) Clonal expansion and reactivation are possible outcomes of the dynamics of latently infected clones upon stimulation by antigens. As a simplifying assumption, susceptible T cells all have identical dynamics in our model, and the dynamics of response to antigen are modeled only for latently infected cells.

not deterministic, follows experimental observations [57,58]. Prior work has also shown that reactivation during homeostatic proliferation is rare [57]. Overall, our model differs from previous ones that considered a constant rate of reactivation for latently infected cells [35] or different cell populations with different half-lives [34]. We simulated our model using a system of stochastic differential equations describing the dynamics of cells during HIV-1 infection both before and during ART (Methods).

**Seeding of the reservoir and clonal proliferation pre-ART.** We first simulated the seeding and development of the latent reservoir during the first months of infection. Simulations begin with a number of virions in the system and zero active and latently infected cells (Methods). We decreased viral infectivity after one month to capture suppression of viral replication by the immune system. Even during the first weeks of infection, we observed a large number of distinct latently infected clones in the reservoir. Most clones are very small: for up to a year after infection, the average clone size is less than 10 cells, with a median clone size of 2 cells (Fig 2). During this time, the largest clone is typically smaller than 1000 cells. Most of these clones are also short-lived. The average age of a clone after one year of infection is 15 days, with a median age of 6 days. Such short-lived clones are highly likely to reactivate, with an average probability of reactivation $p_R$ around 9%.

During this early phase, some clones will be stimulated to proliferate by exposure to antigens. However, the effect on different clones in the reservoir differs substantially depending on how likely the latent virus is to reactivate when stimulated. In clones with low reactivation probability, proliferation due to antigenic stimulation typically results in net growth. In clones that readily reactivate, however, the reactivation of latent virus reduces the effective growth rate due to antigenic stimulation, which can ultimately lead to the elimination of these clones. These dynamics lead to a progressive increase in the number of clones with low reactivation probabilities and large clone sizes over time, consistent with recent work that has observed a progressive decrease in clonal diversity with time [29].

**Kinetics of plasma viral load, HIV-1 DNA, and the inducible viral reservoir after ART initiation.** After the LR has been seeded, we simulated the response of viral populations, including both latent and actively infected cells, to long-term ART. To simulate viral kinetics during ART, we decreased viral infectivity $\beta$ at month 60 due to treatment (Methods), keeping all other parameters constant. Here, we chose 5 years after infection as a start point for ART initiation following typical times from HIV-1 infection to diagnosis [60,61].

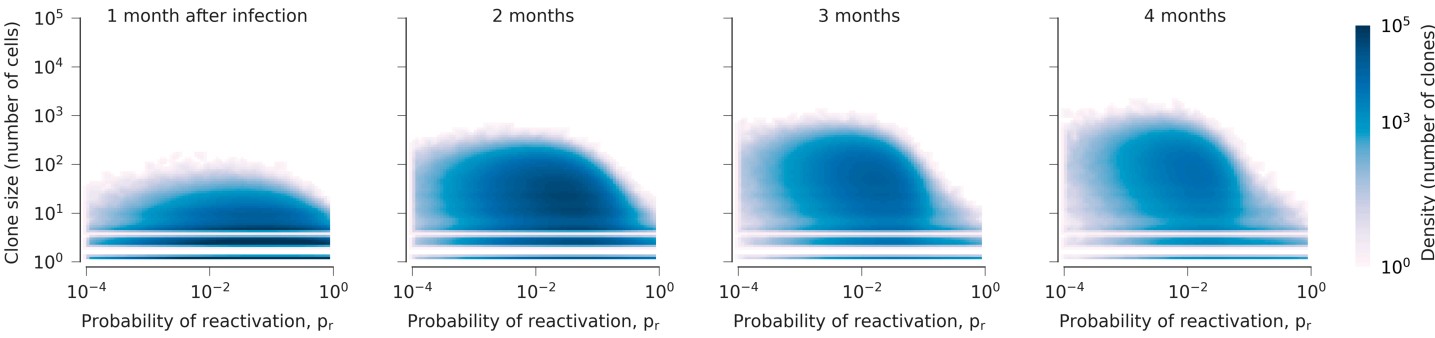

**Fig 2. Distribution of clones in the HIV-1 latent reservoir early in infection.** Most clones are small and have high reactivation probabilities. However, as time passes, clones with lower reactivation probabilities begin to grow in size. Stimulation from antigens drives some rare clones with very low probabilities of reactivation to large sizes. Note gaps in the heatmap occur for clones with <10 cells because we enforce integer numbers of cells for small clones (see Methods).

Clinical data shows that after initiating ART, HIV-1 RNA in blood decreases rapidly over the course of around 2 weeks, with observed half-lives $t_{1/2}^{\text{RNA, early}}$ of 0.9-1.9 days [38,62]. This is followed by a more gradual decline over the next 4 weeks ($t_{1/2}^{\text{RNA, late}} \sim$ 7.8-27.2 days [38,62]). The total number of latently infected cells, measured by HIV-1 DNA in PBMCs, decays steadily over the first few months on ART ($t_{1/2}^{\text{DNA}} \sim$ 99-133 days). Infectious units per million PBMCs (IUPM), measured in viral outgrowth assays (Methods) declines very slowly, with a measured half-life $t_{1/2}^{\text{IUPM}}$ of approximately 44 months [59].

Our model quantitatively recovers the decay rates of HIV-1 RNA and latently infected cells (combined across all clones) spanning days, months, and years on ART (Fig 3; see also S1 Fig). In our simulations, viral load first drops sharply, which is primarily driven by the death of actively infected cells (Fig 3A and S2 Fig). At the same time, small clones are gradually eliminated through reactivation or random cell death. Due to reduced viral replication, these

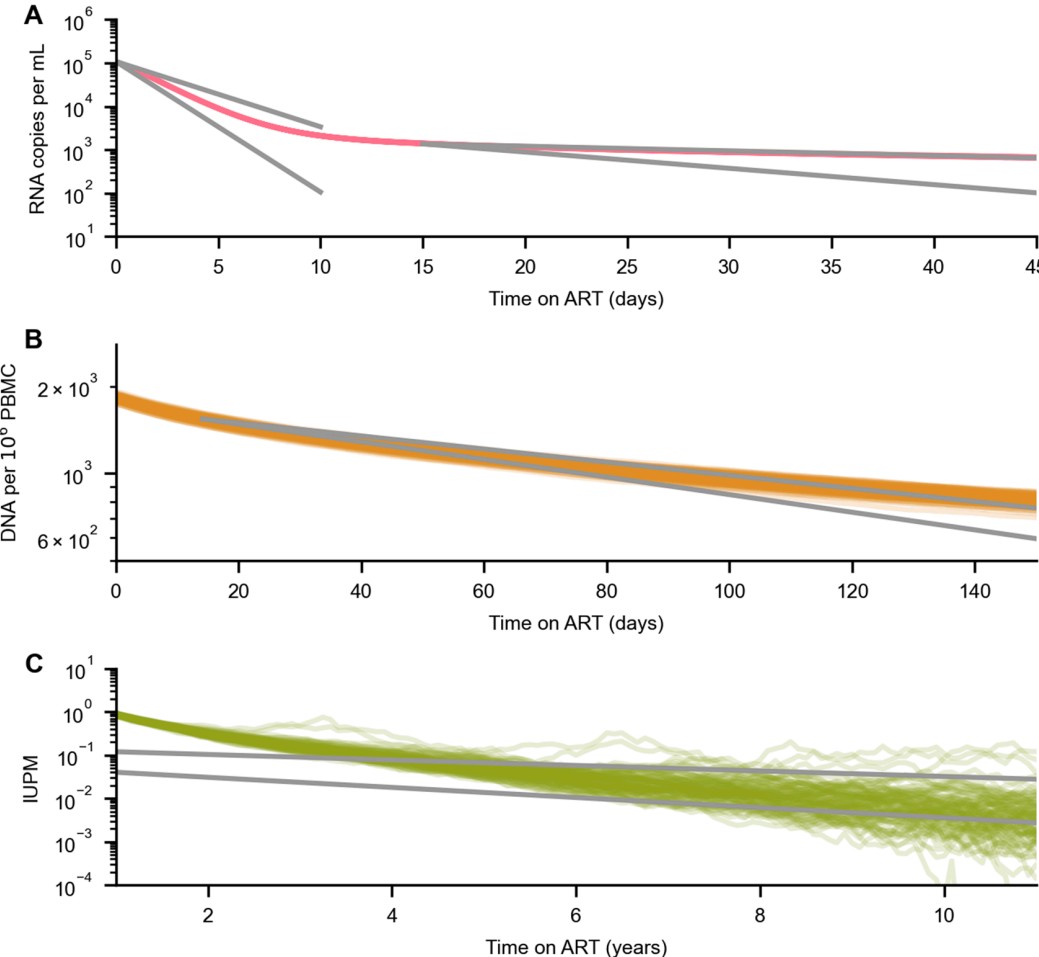

**Fig 3. Latent reservoir dynamics after ART initiation in 100 replicate simulations.** (**A**) ART first results in the rapid decline of plasma viral load [38], first due to the death of actively infected cells and later driven by the elimination of small clones in the LR. (**B**) Over slightly longer times, the number of latently infected cells steadily declines [38] as small clones die out and are no longer quickly replenished by new infections. (**C**) Infectious units per million (IUPM), a measure of cells in the LR capable of reactivation, declines over the course of years, approximately following the 44-month half-life measured in clinical data [59].

clones are no longer replenished at the same rate, leading to a net decline in the total number of latently infected cells. Clones with high reactivation probabilities are depleted more rapidly than those that do not readily reactivate (Fig 4A). Collectively, these factors lead to a shift in the LR toward larger clones with lower rates of reactivation, slowing the decline in viral load and HIV-1 DNA in PBMCs (Figs 3A, 3B, 4A, and S3 Fig). As larger clones are slowly eliminated, we find a decline in the inducible reservoir consistent with measurements from clinical data (Fig 3C).

**Long-term clonal dynamics in the latent reservoir.** During ART, clones with higher probabilities of reactivation have a shorter effective survival time than clones with lower probabilities of reactivation. We therefore find that the average probability of reactivation decreases over time. However, clones that readily reactivate are not entirely eliminated. Occasional reactivation of latent cells from large clones leads to bursts of viral replication that partially reseed the reservoir. These dynamics result in a long-term quasi-steady state, where small clones with high probabilities of reactivation turn over frequently while large, quiescent clones slowly fluctuate in frequency (S4 Fig and S5 Fig).

Over long times, we find that, for the largest clones, clone size $n$ scales inversely with the probability of reactivation $p_r$, $n \propto p_r^{-\alpha}$, with an exponent $\alpha \sim 1$ (Fig 4A). This finding is consistent with previous work that observed a power law relationship between clone size and probability of reactivation in viral outgrowth assays in data from multiple subjects years after ART initiation [25]. Clones that are small and/or have low probabilities of reactivation (i.e., ones occupying the lower left corners in Fig 4A) are particularly challenging to quantify in patient data because their probabilities of being sampled in sequencing HIV-1 DNA from PBMCs or viral outgrowth assays are exceedingly small. Such clones are likely to be observed only once in data if they are sampled, which is consistent with observations in clinical data [25].

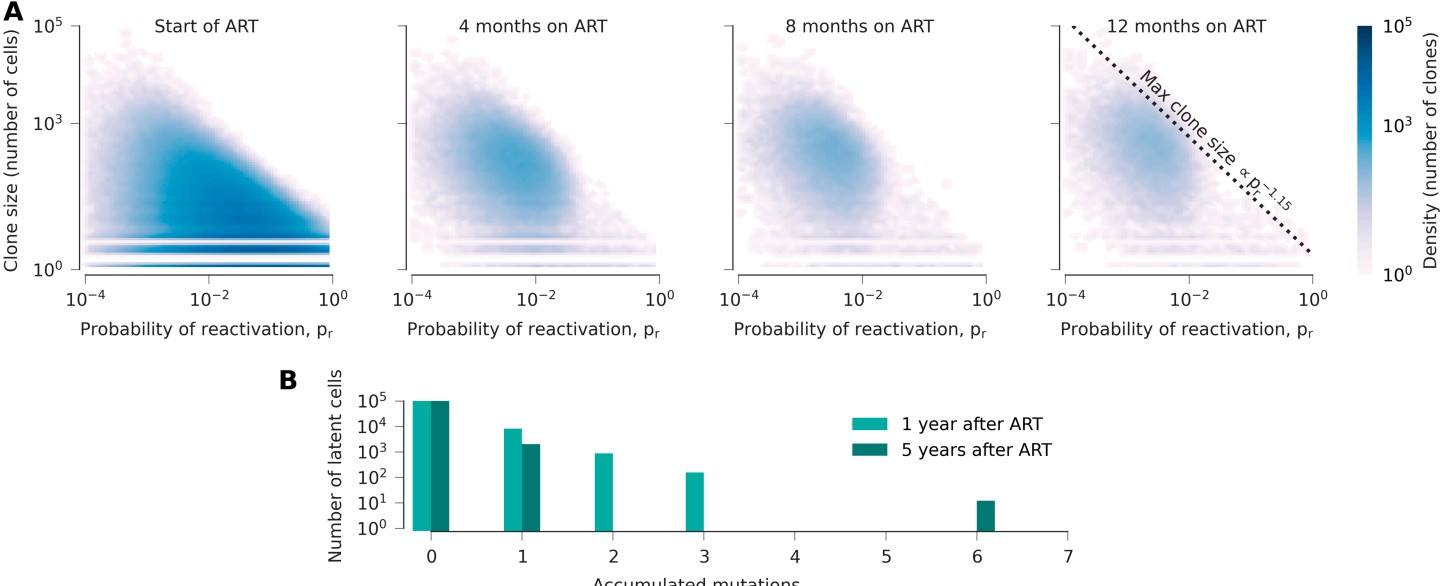

**Fig 4. Dynamics of the latent reservoir during ART.** (**A**) As ART begins, many clones are observed with different sizes and probabilities of reactivation. Reactivation and random fluctuations lead to the preferential loss of small clones and ones with high probabilities of activation over time. The largest clone size scales roughly with the inverse of the reactivation probability, a relationship that is stable over time. (**B**) Despite viral replication, few mutations accumulate in latent clones during ART.

Our results also show that after a year on ART, the relative size of large clones changes little over time (S6 Fig). Collectively, our simulations are consistent with longitudinal studies that found few significant changes in the proportion of different clones in the LR when sequencing proviral DNA [28,63,64], while significant changes in clonal distributions can be observed when sequencing reactivated viruses from viral outgrowth assays [63,64].

**Dynamics of clone age pre- and post-ART.** To further characterize the clonal dynamics of the LR, we tracked clone age (defined as the time since the latent infection event that first established a new clone) and probability of reactivation over time. Before ART begins, most clones in the LR have been recently deposited, though we also observe a spike in the age distribution that corresponds to latent infection events during the early, acute stage of infection (S6 Fig). After ART, the age distribution progressively shifts toward older clones. This is due to the death of small and highly reactive clones during the time immediately after ART initiation, as described above. However, there remains a spike in the age distribution near zero due to new latent infection events in short bursts of viral replication during ART. Overall, this finding agrees with recent work that has found that the decay of younger clones in the LR is faster than for older ones [65,66].

**Processes driving latent cell proliferation and death.** In our model, there are three ways for latent cells to be produced: homeostatic or antigen-driven proliferation of latent clones, and new latent infection events. There are two ways that latent cells can be removed from the reservoir: through reactivation or through homeostatic death. We tracked the contributions of each of these factors to latent cell proliferation and death both before and after ART (S8 Fig). Interestingly, we found that new infection events only constitute the main contribution to latent cell growth during a brief period very early in infection. After the latent reservoir has been seeded, homeostatic proliferation and antigenic stimulation are by far the dominant factors driving latent cell growth, with homeostatic proliferation making the largest contribution in our model. Both of these factors have been cited in prior experimental work as contributors to reservoir growth and persistence [15,16,19–21]. We find that homeostatic death is by far the largest contributor to latent cell death both before and after ART. However, as described above, the uneven probabilities of reactivation across clones result in a gradual shift in the composition of the reservoir (Fig 4A and S3 Fig), even though reactivation constitutes a small fraction of overall latent cell loss.

**Variation in viral dynamics over multiple simulations.** We performed 100 replicate simulations to test the variation in dynamics between different stochastic realizations of the model (Fig 3). Dynamics early after ART initiation were the most repeatable in our simulations, and the dynamics of latent clones after years of ART was the most variable. This is expected because the dynamics immediately after ART initiation involve large numbers of cells and are therefore nearly deterministic, while long-term ART dynamics are driven by fluctuations of smaller numbers of latent clones.

Because our replicate simulations were performed with identical underlying parameters, they demonstrate the minimum level of stochasticity inherent in our model. However, real variation in viral dynamics between individuals are influenced by a broad range of factors, including differences in host immune responses, duration of infection before ART, and therapy adherence and effectiveness, that would best be modeled by variation in simulation parameters. Thus, these replicate simulations represent a lower bound on variation between individuals.

**Choice of reactivation probability distribution and the role of heterogeneous reactivation rates.** Evidence suggests that different latent clones have different propensities for reactivation [22,25,56,67,68]. Ultimately, in our model this heterogeneity in reactivation

plays a central role in concisely explaining widely differing and multiphasic decay rates of HIV-1 RNA in blood, HIV-1 DNA in PBMCs, and IUPM observed in clinical data. Heterogeneous reactivation rates are also essential to reproduce the observed association between the probability of reactivation in viral outgrowth assays and latent clone size [25].

However, there is no singular distribution of reactivation probabilities that uniquely reproduces the observed clinical values. For simplicity, we opted to use a lognormal distribution, which is a natural choice when a variable (i.e., probability of reactivation) is obtained as the product of many independent explanatory variables (e.g., virus genetic background, cell type, integration site, local chromatin context, etc. [68]). Previous modeling work has used a lognormal distribution for turnover rates to define uneven proliferation of clones in the first year of infection [37]. Modeling the probabilities of reactivation with this distribution, which is grounded in the underlying biology of latent infection and consistent with clinical data, allows us to extend our model not only to active infection but also to describe long-term dynamics of the latent reservoir during ART.

Models with reactivation probabilities that were the same for all clones were difficult to fit with constraints on short-term and long-term viral dynamics. With the probability of reactivation set to 1% for all clones, for example, we find that the dynamics shortly after ART begins are recovered well, but the long-term decay of the LR is much faster than expected (S9 Fig).

**Role of heterogeneity in antigenic stimulation.** To gauge the importance of heterogeneous antigenic stimulation in our model, we considered an alternative scenario in which the level of antigenic stimulation was kept at a constant, uniform level across all latent clones. In such simulations, the level of antigenic stimulation must be carefully tuned: low values result in the rapid death of latent clones, while high ones cause unbounded proliferation. Simulations with a concentration of antigen equal to the long-term average in standard simulations successfully recovered viral kinetics immediately post-ART (S10 Fig). However, this condition also led to a more rapid decay of the LR than expected from clinical constraints. In addition, the maximum clone size was sharply limited in simulations with constant antigenic stimulation, more than an order of magnitude smaller than the largest clone sizes in our standard simulations (S10 Fig). This feature is also inconsistent with observations of highly expanded clones that can comprise substantial fractions of the overall intact reservoir within an individual [25,69]. Thus, at least in our model, we find that heterogeneous antigenic stimulation and rates of reactivation are needed to fit the constraints of clinical data.

**Presence or absence of HIV-1 evolution during ART.** While ART strongly suppresses viral replication, it may not be completely effective. Past modeling work has suggested that low amounts of replication can continue after treatment intensification and may influence the level of detectable virus, but are unlikely to allow for long-term sequence evolution [32,34–36]. Others have argued that ongoing HIV-1 replication can lead to measurable viral evolution during ART [70–72], though this point is hotly debated [73,74], and multiple studies have failed to observe evolution in the reservoir during ART [75–78].

To test whether or not viral sequence evolution would occur in our model, we tracked the number of accumulated mutations in individual clones after ART initiation. In our simulations, the mean number of new infection events resulting from active infection in a single cell is smaller than one due to the suppressive effects of ART (S8 Fig). This implies that persistent, self-sustaining active replication is impossible. However, because our model is stochastic, we observe occasional "bursts" of viral replication. In phylogenetic terms, the replication dynamics in our model would thus generate star-like phylogenies (i.e., with few mutations around stable clones in the LR), rather than the ladder-like trees that can be generated from

sequential evolution. Despite occasional bursts, we found no evidence for progressive accumulation of mutations or genetic divergence over time (Fig 4B), consistent with past observations [75–78] and modeling work [32,34–36].

One prominent prior study that argued for continued HIV-1 evolution during ART was based on observations made during the first year of ART [72]. As stated above, we do not find evidence for significant sequence evolution in our model. However, we do observe immense changes in individual clone sizes during early ART, especially for many small clones that are eliminated (S5 Fig). These dynamics support previous arguments that sampling of different clones could explain the *appearance* of evolution shortly following ART [69].

**Effects of early intervention on LR structure.**  Recent studies found that the composition of the LR is altered in individuals who have undergone early ART treatment [79]. To mimic early ART, we adjusted our simulations to include a sharp drop in viral infectivity 15 days after initial HIV-1 infection (Methods). Unlike previous simulations, early intervention results in a much smaller number of clones in the LR (Fig 5A), which quickly becomes

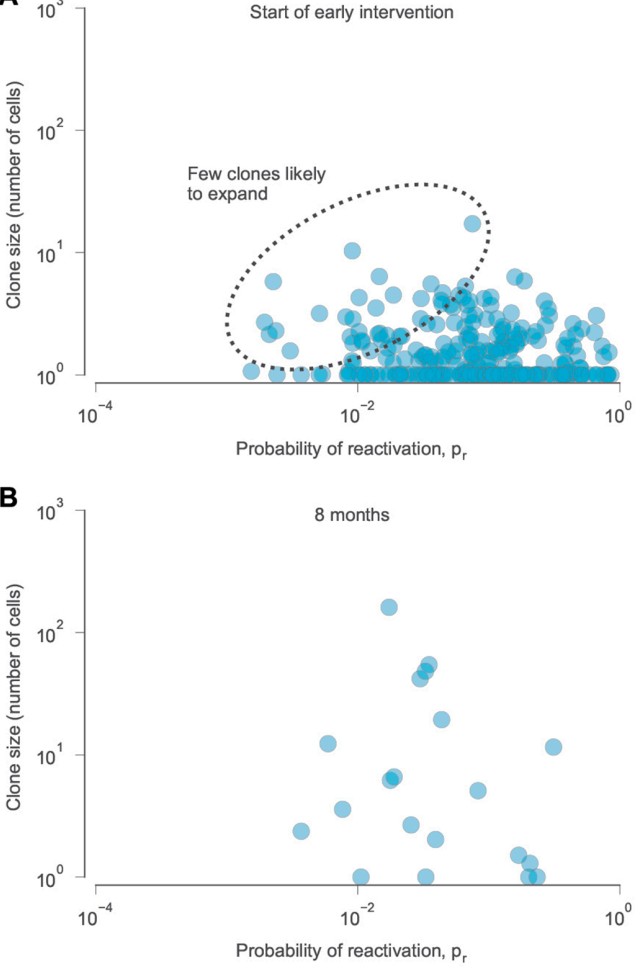

**Fig 5. Distribution of clones in the latent reservoir after early intervention.** (**A**) We simulated the effects of early intervention by initiating ART conditions 12 days after infection. At this time, there are few clones that are large or have small probabilities of reactivation, limiting the potential for clonal expansion. (**B**) After 8 months, few clones remain.

dominated by just a few large clones (Fig 5B). Our simulations thus recapitulate studies showing that the LR in those with early ART are mono- or oligo-clonal, with little reactivation and background replication [79] and fewer intact proviruses [22]. While our model was not calibrated to model analytical treatment interruption (ATI, a process in which individuals temporarily cease ART in a controlled setting), exploratory simulations also showed that viral rebound occurred much slower for the early ART scenario (S11 Fig, Methods). Delayed viral rebound after early ART has been observed both in clinical data [80–82] and in animal models [83,84].

**Factors underlying long-term LR persistence.**   Despite suppressive ART, the latent reservoir persists for decades in HIV-1-infected individuals. How do different components of the LR contribute to its persistence? We find that the presence of large latent clones is the most important factor in the long-term persistence of the latent reservoir, despite their low probabilities of reactivation. In fact, small $p_r$ allows these clones to grow to large sizes with minimal decay due to reactivation (Fig 4A). Because they are large, these clones are also very unlikely to die stochastically due to fluctuations in clone size, unlike smaller clones that turn over rapidly (S5 Fig).

Interestingly, the largest clones are not the ones that are most likely to generate rebound viruses. On average, the rate of viral outgrowth is proportional to the product of the clone size and reactivation probability. In typical simulations, this is maximized by clones with intermediate sizes and reactivation probabilities. This is because clones with very small probabilities of reactivation are fairly rare, and clones with very high probabilities of reactivation tend to be small and short-lived. This can be seen quantitatively by computing the total contribution of clones of different sizes to reactivation during ART (S12 Fig). Thus, even though we find that the largest clones (with small probabilities of reactivation) are chiefly responsible for LR persistence, they are unlikely to be the typical first source of outgrowing viruses during rebound.

Relatedly, it has sometimes been challenging to identify the source of rebound viruses in the latent reservoir in ATI studies [85]. Given that we find clones of size $10^2$-$10^3$ contribute most to reactivation, and considering the size of the LR in our simulations, this suggests that one would typically need to sequence roughly 8,500 intact proviruses to sample a specific clone driving rebound. In the case of early ART, fewer intact proviruses may need to be sampled to identify the rebound virus (roughly 400 in a typical case), given the smaller size of the LR.

## Discussion

Here we developed a stochastic model of the latent reservoir of HIV-1 that accounts for the inherent heterogeneity of different clones in the reservoir. Our mechanistic approach and direct simulation of each clone allows us to delve deeper into the underlying processes shaping reservoir dynamics. Our model recapitulates changes in HIV-1 measurements in clinical data over the scale of days (decline in HIV-1 RNA in the blood after ART) to years (decline in IUPM over years on ART). We also quantitatively recover a "power law" relationship between clone size and reactivation for large clones, and we find realistic distributions of clones in the LR in simulations that mimic early intervention with antiretroviral drug treatment.

Our results complement several recent experimental investigations of the latent reservoir. Studies have shown preferential integration of intact proviruses into transcriptionally silent regions of the genome [17], and that most transcriptionally active proviruses were selected against during ART [18]. Less transcriptionally active proviruses may be in a deeper state of latency, and thus correspond with clones with smaller probabilities of reactivation in our

model. We consistently find that more reactive latent clones are selected against during long-term ART, as reactivation limits their capacity for proliferation and self-renewal. A number of studies have also pointed to the importance of antigenic stimulation in LR persistence [15,21]. In our model, antigenic stimulation is essential for driving the growth of the largest clones; as in prior studies of the T cell repertoire [46], birth-death noise alone is not sufficient to drive very broad differences in clone sizes. Computationally, our work also connects with recent studies that have begun to explore the consequences of heterogeneity and clonal structure in the latent reservoir [37,69,86].

Beyond comparisons with experimental and clinical data, our model makes several predictions about the structure and long-term dynamics of the latent reservoir. Our study suggests that the LR consists of a very large number of clones, especially small clones. Due to their small size and (for some clones) low probability of reactivation when stimulated, they would be difficult to detect through conventional means in studies that seek to characterize the viral reservoir. Nonetheless, collectively, they contribute to the diversity of the latent reservoir and serve as a potential source of viral rebound.

Clonal heterogeneity, including the propensity of different clones for reactivation and stimulation by antigens, emerged as a critical factor to reconcile both short- and long-term dynamics of the latent reservoir after ART. Differences in probabilities of reactivation lead to a slow but progressive "coarsening" of the reservoir, as clones that readily reactivate are eliminated and larger, quiescent ones persist. The persistence of clones with low probabilities of reactivation in our model aligns with recent longitudinal studies that have reported the positive selection of proviruses with lower transcriptional activity during prolonged ART [18].

Our simulations also show nuanced effects of sporadic viral replication during ART. With zero viral replication, all small clones in the LR would ultimately be eliminated due to random clone size fluctuations. However, the level of active replication needed to sustain a population of small clones is insufficient to produce long-term sequence evolution of the virus [32,34–36]. This emphasizes an important distinction between viral replication and evolution, especially evolution within the LR. Persistent, self-sustaining viral replication will lead to the accumulation of mutations (i.e., evolution) over time. However, the same is not true for sporadic bursts of replication that cannot be sustained, and which must be restarted from the same pool of unmutated latent viruses after previous active infections die out. Our model suggests that sporadic replication during ART is consistent with experimental data, but persistent replication is not.

Past work has identified various factors that could affect clonal proliferation, including exposure to antigen and different HIV-1 integration sites [56]. Here, we found that random differences in antigenic stimulation alone are sufficient to reproduce the observed structure of the latent reservoir. This result should not be interpreted as evidence that different integration sites do not play a role in heterogeneous clonal expansion. Rather, our work shows that differences in antigenic stimulation can already lead to stratification in clone sizes and dynamics. Additional factors could also further contribute to the heterogeneity of the LR. For example, a proliferative advantage associated with specific integration sites could promote clonal expansion, potentially extending the lifetime of the reservoir.

Our study has several limitations. In repeat simulations, we chose identical underlying parameters, including the time for ART initiation. In future work, it would be ideal to sample the biologically plausible parameter space more deeply, giving a greater sense of the types of variation in viral dynamics that can or cannot be captured in our model. It could also be helpful to rigorously match simulation conditions with the details of clinical data sets to better constrain model parameters and identify areas where the model must be extended. Our model could also be further tuned to align with outcomes from ATI trials [85] or observed

fluctuations in clone size [87]. Regarding differences in the timing of ART, specifically, we find that the distribution of small clones rapidly reaches quasi-equilibrium soon after acute infection (Fig 2). However, very large clones typically take more time to appear. Thus, we expect viral dynamics that depend on the existence of very large clones (or a very broad distribution of clone sizes) to be most sensitive to the timing of ART initiation.

Extensions to our model could also improve its realism and fit to data. For example, we currently model the effects of host immune responses implicitly by reducing viral infectivity. Explicitly modeling host immunity could allow us to understand how differences in immune responses affect the latent reservoir and its dynamics. This would be useful, for example, in investigating unique properties of the reservoir in elite controllers [88]. In addition, some studies have observed large latent clones that are transcriptionally active [18] and contribute to active viral replication [21]. In our current model, we would expect these instances to be rare. However, it may be possible to explain these cases through differences in the baseline proliferation or death rates of individual clones, independent from antigenic stimulation, as in other recent computational LR models [37].

Many assays have been devised to quantify the magnitude and diversity of the HIV-1 latent reservoir, but quantifying the true size of the LR remains challenging [11]. To address this, a hybrid approach combining stochastic modeling and statistical analysis that accounts for the limitations of experimental noise, similar to proposals for T cell repertoire diversity estimation [89], may offer an effective quantitative measurement of the LR. Our work could contribute to this effort by providing a way to study small clones, which are difficult to access experimentally, using a rigorous model constrained by experimental and clinical data.

Understanding the structure and dynamics of the HIV-1 latent reservoir could aid in the development of HIV-1 cure strategies that aim to eliminate or permanently suppress the LR. Our model contributes to these efforts by providing a quantitative description of the LR that is consistent with existing data, but which also extends to "unseen" areas that are difficult to characterize experimentally. One important finding relevant for HIV-1 cure strategies is that large clones that are replication-competent but relatively unlikely to reactivate play a key role in long-term persistence of the LR. In future work, our model could be used to simulate responses to different types of therapeutic interventions, evaluating plausible paths to an HIV-1 cure.

## Methods

Our model consists of four main populations: uninfected CD4$^+$ T cells (T), productively infected activated CD4$^+$ T cells (A), latently infected resting CD4$^+$ T cells (L), and HIV-1 virions (V). All cells have finite lifespans determined by their respective death rates, and virions are cleared at a clearance rate $c$. Uninfected cells are produced at a constant, fixed rate, and virions at a rate proportional to the number of active cells. HIV-1 virions can infect target CD4$^+$ T cells, and upon infection, a small fraction will result in cells with defective proviruses due to effects like large deletions or hyper-mutations. Functional proviruses will accumulate a mutation with probability $p_{mut}$. Finally, a small fraction of infection events will result in latently infected cells. To account for stochasticity, these processes are modeled with probabilities instead of rates.

To account for the heterogeneity of the LR in our model, we define a latently infected clone as a set of cells that have the same TCR, proviral DNA sequence, and integration site. Individual clones will differ in how they are stimulated by antigens and their propensity for reactivation. Due to differences in integration sites, each new clone $L_i$ is assigned a random probability of reactivation $p_{r_i}$. Following work describing dynamics of T cell repertoires [46],

we describe the stimulation of each clone by antigens with $f_i(t) = \sum_{j=1}^{m} k_{ij} a_j(t)$ where $k_{ij}$ is the interaction coefficient between clone $i$ and antigen $j$ (when clones are cross-reactive), and $a_j(t)$ is the overall concentration of an antigen $j$ as a function of time. We assume that antigen concentration decays exponentially after its introduction at random times as pathogens are encountered and cleared, either passively or through the action of the immune response.

When a latently infected cell is stimulated to divide, there is a probability of the latent provirus reactivating, converting the cell into an actively infected cell. In this case, the number of latent cells decreases by 1 and the number of actively infected cells increases by 1. If no reactivation occurs, then the latently infected cell proceeds to divide in response to the antigen interaction.

The dynamics followed by a latently infected clone $L_i$ are then driven by its basal division rate $\nu_L$, death rate $\mu_L$, probability of reactivation $p_{r_i}$, and its interaction with antigens $f_i(t)$. These dynamics can be described by a stochastic process in which events (division, death, ...) are selected to occur with probabilities proportional to their rates. After each event, time then advances by a random increment that is exponentially distributed following the sum of all of the rates [90].

While mathematically exact, this explicit simulation approach becomes extremely computationally intensive when dealing with large systems (i.e., large numbers of cells or virions, in our case). However, in this limit the dynamics can be simplified. One can instead then describe the evolution of different populations according to stochastic differential equations (SDEs), which feature a deterministic component and a random term that adds noise to the dynamics [90]. This noise can represent both finite population noise (i.e., the system size is not infinite, so by chance there may be slightly more or less than the expected number of cell deaths at some point in time, for example) and stochastic variation in the environment (i.e., fluctuations in the concentration of antigens, in our case). As one prototypical example, Brownian motion can be modeled with SDEs.

Taking this limit, the dynamics of each latent clone $i$ are described by the SDE

$$dL_i = \left[ \left( 1 - 2p_{r_i} \right) f_i + \nu_L - \mu_L \right] L_i dt + \sqrt{\left( \nu_L + f_i + \mu_L \right) L_i} \, dW_{L_i} \tag{1}$$

Here, the first term multiplying the time step $dt$ is the deterministic term, representing changes in latent clone size from antigenic stimulation and reactivation, homeostatic proliferation, and death, respectively. The second term multiplying the Gaussian white noise, $dW_{L_i}$, quantifies the random fluctuations in these contributions to clone size fluctuations.

As mentioned above, the function $f_i(t)$ encodes the fluctuating level of antigenic stimulation experienced by clone $i$. The stochastic process giving rise to $f_i(t)$ is a sum of Poisson-distributed, exponentially decaying spikes. This process is not easily amenable to analytical treatment or simulations, so following the approach of Desponds et al. [46], we assume that correlations among clones are weak and replace the function with a simpler one with the same temporal autocorrelation, that is an Ornstein-Uhlenbeck process:

$$\frac{df_i}{dt} = -\lambda_f f_i + \sqrt{2} \gamma_f \eta_i(t) . \tag{2}$$

Here $\eta_i(t)$ is a Gaussian white noise, $\lambda_f$ is the inverse of the characteristic lifetime of antigens, and $\gamma_f$ quantifies the strength of variability of the antigenic environment. The antigen concentration experienced by each clone, then, stochastically fluctuates around a baseline value over the course of the simulation.

To model the active cells and virions we need to consider what happens during an infection event, which is illustrated in Fig 1. A virion with sequence $k$ finds and successfully infects a susceptible T cell at rate $\beta$. During infection, there is a probability $p_{\text{def}}$ that the integrated provirus will be defective due to large deletions, hypermutation, or other similar alterations. For proviruses that are not defective, we model the accumulation of point mutations with probability $p_{\text{mut}}$. Finally, we consider a probability $p_L$ for the infection to be latent. Each latent infection defines a new latent clone, since we assume that the probability that two identical viruses integrate at the same location in two T cells with identical T cell receptors in separate infection events is essentially zero. This clone could share the same sequence as another latently infected clone but have a very different integration site and, therefore, a different probability of reactivation.

In the same limit as above, the dynamics governing actively infected cells and virions are defined by the following system of SDEs:

$$\begin{bmatrix} dA_k \\ dV_k \end{bmatrix} = \begin{bmatrix} \sum_{i=1}^{n} f_i p_{r_i} L_i + \left(1 - p_{def}\right)\left(1 - p_{mut}\right)\left(1 - p_L\right)\beta T V_k - \mu_A A_k \\ \gamma A_k - \beta T V_k - c V_k \end{bmatrix} dt + D dW,$$ (3)

$$DD^T = B,$$ (4)

$$B = \begin{bmatrix} \sum_{i=1}^{n} f_i p_{r_i} L_i + \left(1 - p_{\text{def}}\right)\left(1 - p_{\text{mut}}\right)\left(1 - p_L\right)\beta T V_k + \mu_A A_k & -\left(1 - p_{\text{def}}\right)\left(1 - p_{\text{mut}}\right)\left(1 - p_L\right)\beta T V_k \\ -\left(1 - p_{\text{def}}\right)\left(1 - p_{\text{mut}}\right)\left(1 - p_L\right)\beta T V_k & \gamma A_k + \beta T V_k + c V_k \end{bmatrix}.$$ (5)

In other words, change in the number of actively infected cells with sequence $k$ is driven by: 1) latent reactivation, 2) new infection events that do not result in a defective virus or latent infection (mutants have a different sequence $k'$, and thus they also do not contribute here), and 3) the death of actively infected cells. Virions are produced at what we assume for simplicity to be a constant rate from actively infected cells, and they can be lost either from clearance/degradation by the host or through new infection events. The noise term $D$ couples the fluctuations of the virions and actively infected cells.

Finally, the susceptible T cells in our model follow the simple stochastic differential equation

$$dT = \lambda_T dt - \beta T V dt - \mu_T T dt + \sqrt{(\lambda_T + \beta T V + \mu_T T)} dW_T.$$ (6)

That is, new susceptible T cells are produced at a constant rate by the thymus, and they are lost by either infection (in which case they become actively infected cells) or death.

## Tuning the reactivation probability distribution

We explored various values of $\mu$, the average of the logarithm of probabilities of reactivation. We then adjusted $\sigma$, the spread of the logarithm of probabilities of reactivation to align with the multiphasic decay patterns observed in viral load, HIV-1 DNA, and IUPM measurements. Generally, as $\mu$ increases, the number of clones with high probabilities of reactivation and smaller sizes increases. This leads to a sharper drop in HIV-1 DNA post ART and a shorter

time for the reservoir to become oligo-/monoclonal. Thus, the values of $\mu$ and $\sigma$ are constrained, if not completely determined, by existing clinical data. To illustrate our findings, we used a log-normal distribution with $\mu = -1$ and $\sigma = 0.8$. Experimental measurements of the distribution of probabilities of reactivation would be of great interest, allowing us to more precisely model long-term behavior of the reservoir.

## Modeling seeding of the reservoir and ART initiation

To simulate the initial establishment of the reservoir, we first calibrated the parameter $\beta$ to capture the observed rise in viral load in the early stages of infection [91,92]. Subsequently, after a month, we decreased the value of $\beta$ to emulate the immune system's suppression of viral replication, while maintaining fixed conditions for clonal proliferation prior to ART initiation. This adjusted $\beta$ value determines the viral load set point in chronic infection, and these conditions remain constant until the initiation of ART, which in our primary example simulation occurs 60 months post-infection.

Upon ART initiation, the impact of treatment is simulated by modifying the value of $\beta$. Specifically, this adjustment aims to align the initial decline in viral load in simulations with the decay of HIV-1 RNA in blood observed in clinical data during the first two weeks following ART initiation [38]. Given that the number of virions we observe is proportional to the number of active cells and to the viral load, we follow the methodology outlined in Hill et al. [44] to estimate viral load, where the number of actively infected cells is divided by 1680. This value is the geometric mean of different estimates from clinical data for the cell to virus ratio, obtained by balancing viral production and decay at equilibrium with an estimate that virus particles in the lymphoid tissue outnumber the ones in circulation 100-fold. Throughout the pre-ART and ART periods, all other parameters are held constant and remain unchanged.

To replicate scenarios involving elite control or early ART initiation, we introduced a rapid decline in infectivity ($\beta$) shortly after the initial HIV-1 infection. Rather than waiting for 60 months to commence ART, we transitioned to the $\beta_{ART}$ value after only half a month of infection.

## Metrics for quantifying the HIV-1 latent reservoir and infection

We used infectious units per million (IUPM) to quantify the abundance of replication-competent HIV-1, which is measured in viral outgrowth assays. In our simulations, we used the product of each non-defective clone's size and its probability of reactivation, summed over all clones, as a proxy for IUPM. This quantity should indeed be proportional to the probability that a latent clone is sampled and successfully reactivates when stimulated, which is analogous to IUPM.

We quantified HIV-1 DNA per $10^6$ peripheral blood mononuclear cells (PBMCs) by dividing the total number of latent cells by the total number of T cells and multiplying the result by $10^6$.

As described above, we quantified HIV-1 RNA in blood in our simulations by dividing the current number of actively infected cells by 1680, the geometric mean of different estimates for the cell to virus ratio, obtained by balancing viral production and decay at equilibrium with an estimate that virus particles in the lymphoid tissue outnumber the ones in circulation 100-fold [44].

## Computational implementation

We used the Euler-Maruyama method to simulate the dynamics of our system of stochastic differential equations (SDEs) [93]. This numerical technique allows us to approximate the deterministic component of the SDEs using the Euler method at each time step. To incorporate the stochastic component, a random term is introduced, generated by a normally distributed random number with a mean of zero. The standard deviation of this random term was determined by the coefficients present in the SDEs.

Due to the complexity of our model, it was not computationally feasible to simulate the full model using a realistic number of CD4$^+$ T cells, roughly $1.75 \times 10^{11}$ for a typical adult [94]. We therefore used two complementary approaches to perform our simulations. First, we simulated the full model at smaller system sizes (i.e., numbers of CD4$^+$ T cells). Second, we developed and simulated simplified models that could readily scale to larger system sizes. As described in sections below, we carefully compared the output of both the full and simplified models for smaller system sizes to ensure that the simplified models accurately captured LR dynamics from the full model.

Here we refer to the order of magnitude of a simulation as the total number of CD4$^+$ T cells included in the simulation. For example, we refer to a simulation including a realistic number of CD4$^+$ T cells for an adult, around $1.75 \times 10^{11}$, as a simulation at order 11. The figures presented in the full simulation were based on an order of magnitude of 9, that is, a total number of CD4$^+$ T cells of $1.75 \times 10^{7}$. In these simulations, the thymic production of T cells, viral infectivity, and metrics for quantifying HIV-1 presence and the LR are scaled in proportion to the total number of T cells in the simulation. For example, if the total number of CD4$^+$ T cells decreases by an order of magnitude, the infectivity $\beta$ increases an order of magnitude such that the product $\beta T$ remains the same.

## Simulation of analytical treatment interruption

We performed initial simulations to mimic analytical treatment interruption (ATI) studies, which stop ART treatment after a certain period of time. Here, we defined rebound as the point where viral load exceeds the detectable threshold of 200 RNA copies per mL. We calculated the rebound time by taking the difference (in days) between when this threshold was reached and when ART was interrupted. In our simulations, we mimicked ART interruption by changing the infectivity parameter $\beta$ from its value during ART, $\beta_{ART}$, to its value during chronic infection, $\beta_{AI}$. However, we note that our model parameters were not adjusted to reproduce average statistics from ATI trials. In future work, such data could be used to further refine the model.

## Supporting information

**S1 Table. Table of parameters used in the model.** The infectivity parameters are listed for an order 11 simulation, but should be increased by one order of magnitude for every one order decrease in the simulation.
(PDF)

**S1 Fig. Viral kinetics in a typical simulation at order 10.** Dynamics of virions, latently infected cells, and actively infected cells for the first ten years of infection in an order 10 simulation. Viral infectivity drops one month after infection to mimic partial immune control, and ART begins five years after infection.
(EPS)

**S2 Fig. Relative contributions to actively infected cells.** In our model, two processes can create actively infected cells: new infection events and the reactivation of latent viruses. Here we show the fraction that each process contributes to the production of actively infected cells over the course of a typical simulation. Before ART, new infection events are by far the dominant contribution to active infections, except for a short time after acute infection when infectivity drops significantly, modeling partial immune control of viral replication. Once ART begins, most actively infected cells arise through the reactivation of latents rather than new infections. This means that viral replication during ART occurs in self-limiting bursts: reactivation events produce less than one new active infection, on average.
(EPS)

**S3 Fig. Distribution of latent clone reactivation probabilities pre- and post-ART.** Before ART, the distribution of probabilities of reactivation for clones in the latent reservoir mostly follows the underlying probability distribution used in our simulations (Methods), indicating little selection for or against reactivation during this time. After ART, the distribution progressively shifts towards smaller probabilities of reactivation as more reactive clones are purged from the reservoir.
(TIF)

**S4 Fig. Distribution of clones in the HIV-1 latent reservoir in a typical simulation.** Before ART, clones are broadly distributed in size and reactivation probability. After ART, small clones lost to stochastic fluctuations are no longer completely replenished through new infections. Clones with higher probabilities of reactivation also preferentially eliminated.
(TIF)

**S5 Fig. Distribution of clone ages and sizes in a typical simulation.** (**A**) Distribution of ages (in months) for large clones with low probabilities of reactivation ($n > 100$ cells and $p_r < 0.01$) and small clones with high probabilities of reactivation ($n < 100$ cells and $p_r > 0.01$) in one simulation. (**B**) Distribution of clone sizes from the start of ART until 3 months after ART initiation, showing the rapid elimination of small clones after ART begins.
(TIF)

**S6 Fig. Fold change in clone size for large clones after ART.** Here we show the distribution of ratios of clone size at two and four years after ART initiation and clone size at one year after ART initiation, specifically for large clones ($n > 1000$ at one year after ART). On average, clones decrease slowly in size over time.
(TIF)

**S7 Fig. Distribution of latent clone ages pre- and post-ART.** Before ART begins, most clones are young, though there is also a significant spike in the distribution corresponding to clones produced during the early, acute phase of infection. After ART, the clone age distribution shifts toward larger values, with substantial contributions from clones deposited very early in infection.
(TIF)

**S8 Fig. Contribution of different processes to latent cell proliferation and death.** (**A**) Relative contribution of antigenic stimulation, homeostatic proliferation, and new infection events towards LR growth. New infection events are only dominant in the early, acute phase of infection. (**B**) Relative contribution of reactivation and homeostatic death towards LR decay. While reactivation shapes the long-term composition of the LR (S3 Fig), homeostatic death is largely responsible for the reduction in the overall size of the reservoir.
(EPS)

**S9 Fig. Distribution of clone sizes and post-ART viral kinetics in a typical simulation with a constant probability of reactivation for all clones.** The inducible reservoir decays rapidly in a simulation with the probability of reactivation set to 1% for all clones.
(EPS)

**S10 Fig. Distribution of clone sizes and post-ART viral kinetics in a typical simulation with a constant level of antigenic stimulation for all clones.** Compared to standard simulations, the maximum clone size is limited and the inducible reservoir decays more rapidly.
(EPS)

**S11 Fig. Time to rebound in example simulations of ART interruption.** We performed exploratory simulations mimicking analytical treatment interruption (ATI)—a process in which individuals temporarily cease ART in a controlled setting—by restoring the infectivity parameter $\beta$ to its setpoint value after some time on ART. We defined time to rebound as the first time with viral load >200 copies/mL. (**A**) In standard simulations, the typical time to rebound after five years on ART was 10 days. (**B**) For early ART simulations, time to rebound could be much more variable due to the small size of the reservoir. Thus, we simulated ATI at one year post-ART in the early ART case to obtain a tighter distribution in times to rebound. Here, the mean time to rebound for early ART was 26 days, substantially longer than the time to rebound in the standard simulations despite a shorter time on ART.
(TIF)

**S12 Fig. Contributions to reactivation from latent clones of different sizes during ART.** Small to medium-sized clones (roughly between 10 and 1000 cells) contribute more to reactivation during ART than larger ones, despite their longer persistence.
(EPS)

**S13 Fig. Comparison of decays between full simulation and simplified model.** Decays of the total number of latently infected cells as well as the rate of new latent clones being produced. (**A**), Start of ART. (**B**), First year on ART.
(EPS)

## Author contributions

**Conceptualization:** Marco Garcia Noceda, John P. Barton.

**Data curation:** Marco Garcia Noceda.

**Formal analysis:** Gargi Kher, Shikhar Uttam.

**Investigation:** Marco Garcia Noceda, Gargi Kher, Shikhar Uttam, John P. Barton.

**Methodology:** Marco Garcia Noceda, Gargi Kher, Shikhar Uttam, John P. Barton.

**Project administration:** John P. Barton.

**Software:** Marco Garcia Noceda, Gargi Kher.

**Supervision:** Shikhar Uttam, John P. Barton.

**Validation:** Gargi Kher.

**Visualization:** Marco Garcia Noceda, Gargi Kher.

**Writing – original draft:** Marco Garcia Noceda, John P. Barton.

**Writing – review & editing:** Marco Garcia Noceda, Gargi Kher, John P. Barton.

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
