## [Decision Letter · Decision Letter 0]

25 Nov 2024

PCOMPBIOL-D-24-01812Clonal heterogeneity and antigenic stimulation shape persistence of the latent reservoir of HIVPLOS Computational Biology Dear Dr. Barton, Thank you for submitting your manuscript to PLOS Computational Biology. After careful consideration, we feel that it has merit but does not fully meet PLOS Computational Biology's publication criteria as it currently stands. Therefore, we invite you to submit a revised version of the manuscript that addresses the points raised during the review process. The reviewers made thoughtful and detailed comments on your manuscript, so please carefully consider their comments and provide detailed answers, including changes in the manuscript. These answers and corresponding changes will be critical in further decisions by the Editorial team. Please submit your revised manuscript within 60 days Jan 25 2025 11:59PM. If you will need more time than this to complete your revisions, please reply to this message or contact the journal office at ploscompbiol@plos.org. Please include the following items when submitting your revised manuscript: * A rebuttal letter that responds to each point raised by the editor and reviewer(s). You should upload this letter as a separate file labeled 'Response to Reviewers'. This file does not need to include responses to formatting updates and technical items listed in the 'Journal Requirements' section below. * A marked-up copy of your manuscript that highlights changes made to the original version. You should upload this as a separate file labeled 'Revised Manuscript with Track Changes'. * An unmarked version of your revised paper without tracked changes. You should upload this as a separate file labeled 'Manuscript'. If you would like to make changes to your financial disclosure, competing interests statement, or data availability statement, please make these updates within the submission form at the time of resubmission. Guidelines for resubmitting your figure files are available below the reviewer comments at the end of this letter. We look forward to receiving your revised manuscript. Kind regards, Ruy M. RibeiroAcademic EditorPLOS Computational Biology Amber SmithSection EditorPLOS Computational Biology Feilim Mac GabhannEditor-in-ChiefPLOS Computational Biology Jason PapinEditor-in-ChiefPLOS Computational Biology **Journal Requirements:**

At this stage, the following Authors/Authors require contributions: John P Barton. Please ensure that the full contributions of each author are acknowledged in the "Add/Edit/Remove Authors" section of our submission form.

3) Please provide an Author Summary. This should appear in your manuscript between the Abstract (if applicable) and the Introduction, and should be 150u2013200 words long. The aim should be to make your findings accessible to a wide audience that includes both scientists and non-scientists. Sample summaries can be found on our website under Submission Guidelines:

4) Please include the heading "Abstract" in the manuscript. Please ensure all required sections are present and in the correct order. Make sure section heading levels are clearly indicated in the manuscript text, and limit sub-sections to 3 heading levels. An outline of the required sections can be consulted in our submission guidelines here:

5) Please upload all main figures as separate Figure files in .tif or .eps format. For more information about how to convert and format your figure files please see our guidelines: 

6) Please ensure that all table files have corresponding citations within the manuscript. We noticed that table 1 is not cited within the text. Please include the in-text citation of the table in the manuscript.

7) We have noticed that you have uploaded Supporting Information files, but you have not included a list of legends. Please add a full list of legends for your Supporting Information files after the references list.

8) We notice that your supplementary Figures are included in the manuscript file. Please remove them and upload them with the file type 'Supporting Information'. Please ensure that each Supporting Information file has a legend listed in the manuscript after the references list.

9) Thank you for uploading your study's underlying data set. We notice that there is a GNU GENERAL PUBLIC LICENSE on your data. We would encourage you to consider using a license that is no more restrictive than CC BY, in line with PLOS’ recommendation on licensing (http://journals.plos.org/plosone/s/licenses-and-copyright). For a list of recommended repositories and additional information on PLOS standards for data deposition, please see [insert relevant link] https://journals.plos.org/ploscompbiol/s/recommended-repositories

**Reviewers' comments:**Reviewer's Responses to Questions

**Comments to the Authors:**

**Please note the review is uploaded as an attachment.**

Reviewer #1: review attached

Reviewer #2: The manuscript builds a stochastic differential equation model of HIV infection, involving HIV-1 virions, uninfected CD4+ T cells, productively infected activated CD4+ T cells, and latently infected resting CD4+ T cells. Clones of CD4+ T cells are defined as latently infected cells with identical T cell receptors, integrated proviruses and integration sites. When exposed to their corresponding antigen, a clone may reactivate with a certain probability. If reactivation does not occur, exposure to the antigen results in clonal expansion. This dynamic between antigen stimulation and clonal expansion leads to several interesting predictions regarding the dynamics of the latent reservoir, including a progressive increase in the number of clones with low reactivation probabilities and large clone sizes over time.

The authors have carried out the simulations with a realistic number of CD4+ T cells where possible, and made clever simplifications where not possible. The simulations agree quantitatively with experimentally observed dynamics of HIV-1 days after the initiation of ART, and IUPM years after the initiation of ART. The model also dives into the possibility of HIV replication and/or evolution in the LR, and suggests that sporadic replication during ART is consistent with experimental data, but persistent replication is not.

Overall, the paper is well written, and presents a compelling case for the relationship between clonal heterogeneity and antigen stimulation, and the dynamics of the latent reservoir. However, a few questions remain, that need to be addressed.

Questions

The authors state that the model of clonal proliferation in response to antigen stimulation implemented here differs from previous studies. However, the biological rationale or justification for the choices made here seems to be missing. It would be great if the biological rationale can be provided explicitly, along with relevant references.

The model description in the methods seems to skip several steps between the explanation and the equations. In fact, some of the variables don’t seem to have explanations at all, such as W_(L_i) in equation (1), D in equation (3). It would be great if all the steps could be worked out, at least in the supplement.

Paragraph 3 of the introduction contains several references (refs 15 to 22) in which the findings are relevant to Figures 2, 4, and 5. It would be good to compare the model predictions with the data in these papers, either directly as has been done in Figure 3, or at least in the discussion.

Cho et al. 2022 [1] found that the clonality of the intact latent reservoir increases and its diversity decreases over time. It would be good to cite this paper and compare and/or comment.

In the results presented in the main manuscript, the time of ART initiation has not been mentioned, except in the results relating to early intervention or elite control. The Methods section states that ART initiation occurred in the simulations at 60 months post-infection.

It would be good to state the time of ART initiation in the simulations explicitly in the results section.

A rationale needs to be provided for picking 60 months post-infection as the time for ART initiation.

Figure 3 includes constraints from clinical data, and the simulation results are generated for the scenario when ART was initiated 60 months post-infection. This needs to be compared with the time when ART was initiated in the clinical data.

It would be good to comment on how the time of ART initiation may affect the phenomena being modeled, and the dynamics of the latent reservoir.

Minor questions

In the first paragraph in the Mathematical model (Methods), the text says “HIV-1 virions can infect target CD4+ T cells, and upon infection, a small fraction will result in defective cells due to …”. I believe the authors meant defective proviruses, right?

The section “Modeling seeding of the reservoir and ART initiation” mentions Fiebig stages 1, 2, 4 and 5. What about stage 3?

References

[1] Cho, Alice, et al. "Longitudinal clonal dynamics of HIV-1 latent reservoirs measured by combination quadruplex polymerase chain reaction and sequencing." Proceedings of the National Academy of Sciences 119.4 (2022): e2117630119.

Reviewer #3: Noceda and Barton have put together a beautiful modeling study that is the first to explore a fundamental component of HIV latent reservoir biology, that there is a continuum of proliferation and reactivation rates/probabilities across various sequence clonotypes. This is an excellent contribution and helps inform how the reservoir persists as well as why we see the type of data we see in experiments.

Sincerely,

Daniel Reeves, PhD

I had a few larger questions/comments:

1) The manuscript covers a lot of ground in a relatively short work, I do think it could be advantangeous to focus the paper slightly and come to more concrete conclusions in several places. Just as a couple examples, it isn't clear exactly what features of the experimental data are missed in a model without clonal heterogeneity (maybe something like Fig S1?), and it isn't clear what fraction of latent cells you expect are generated via replication vs Ag stimulation vs homeostatic proliferation and what fraction are cleared via homeostatic death vs reactivation. I think writing a bit more like this would make the paper more enticing to experimentalists in this field.

2) It seems biologically plausible that proliferation and reactivation are linked together, which if I understand correctly is not considered in this version of the model? It could be important to show whether or not this interaction changes any conclusions (I guess somehow correlating the draws of p_r_i and f_i?)

3) It's been very hard to find the clones that eventually actually rebound in ATI studies, could you possibly do a rough calculation here for how many cells one would need to sample to start finding those rarer/reactivation competent proviruses in a person on ART for years (started early vs late)?

Minor comments:

-when you say genetic material incorporated... you could be a little more precise and say something like viral RNA is reverse transcribed and resulting DNA integrated into the cell's genome, also would give a chance to define provirus here

-would maybe restate "many remain capable" because the majority are defective, and later you mention the extra rarity of reactivation competent cells

-double check whether those refs 9-11 say 1 in 1e4 T cells vs CD4+ T cells?

-the para on biological mechanisms might be expanded just a little to make sure a reader gets that there are a bunch of possible mechanisms operating, from homeostasis (cite Chomont), Ag proliferation (TCR linked, cite Lillie Cohn and Simonetti papers), longevity (cite Lisa Frenkel) and integration site transcriptional effects (probably should cite wagner/maldarelli Science 2014 as well as newer work on silencing from Matthias Lichterfeld/Xu Yu among others here)

-similarly, it would be good to write a little more to introduce the system you are trying to model: a multi-clonal population where certain HIV seq clones are integrated into certain TCR seq clones

-fig1, love this figure, but one part that isn't clear visually is whether the susceptible cells are clonal (I think they are? and power law unevenly at that)

-minor bookkeeping but the early Perelson HIV models didn't really have latent compartments, maybe cite Jessica Conway / Alison Hill and our group if you like

-"reactivation dominates, suppressing" this makes it seem like the reactivation does something to proliferaiton? but you just mean that the reactivation is more common so net is negative right?

-might need to explain that the levels in fig3 are the sum of all the clones

-I would love to see the clonal dynamics during fig3, it would be a super nice visual to color clones by reactivation rate and show the diversity decreasing but also the trend toward large but stable defective clones over time after ART

-one thing Matthias Lictherfeld's group has shown also is that intact sequences can persist for a really long time if they are transcriptionally silenced (ie in specific integration sites). I think your model could be framed to explain this: occasional intact sequences that have very low reactivation probability -- in the model now this is assumed to be about Ag stimulation, but could encompass the other mechanism if you want to write about that in discussion

-fig4B on reservoir seeding, maybe you could frame in terms of what you see is bursts not leading to "sequential" evolution, i.e. star-like vs ladder-like

-relatedly, does your model make any predictions about the frequency of blips and unsuppressable viremia?

-"mimic elite control or early ART" - two points on this: 1) I think EC is a lot more complicated than is handled here so it might be better to just focus on early ART, 2) to go a little further, it would be interesting to see if your simulated reservoirs with early ART also woudl rebound slower? I think there's something interesting going on here with the balance between reactivation probability but also few clones/small reservoir

-we've done some recent work to try to model both the HIV clones and the TCR clones that harbor them (https://www.medrxiv.org/content/10.1101/2024.02.13.24302704v1) it would be very nice to see your formulation extended eventually to see if it already captures the carrier cell dynamics too

-there's also a nice paper on TCR clone modeling that might be contextualized/contrasted with your model and results: https://www.pnas.org/doi/10.1073/pnas.2207516120

**Have the authors made all data and (if applicable) computational code underlying the findings in their manuscript fully available?**

Reviewer #1: Yes

Reviewer #2: Yes

Reviewer #3: Yes

PLOS authors have the option to publish the peer review history of their article (what does this mean?). If published, this will include your full peer review and any attached files.

Reviewer #1: No

Reviewer #2: No

Reviewer #3: **Yes: **Daniel Reeves

**Figure resubmission:** While revising your submission, please upload your figure files to the Preflight Analysis and Conversion Engine (PACE) digital diagnostic tool, https://pacev2.apexcovantage.com/. PACE helps ensure that figures meet PLOS requirements. To use PACE, you must first register as a user. Registration is free. Then, login and navigate to the UPLOAD tab, where you will find detailed instructions on how to use the tool. If you encounter any issues or have any questions when using PACE, please email PLOS at figures@plos.org. Please note that Supporting Information files do not need this step. If there are other versions of figure files still present in your submission file inventory at resubmission, please replace them with the PACE-processed versions.
---

## [Decision Letter · Decision Letter 1]

13 Aug 2025

Dear Dr Barton,

We are pleased to inform you that your manuscript 'Clonal heterogeneity and antigenic stimulation shape persistence of the latent reservoir of HIV' has been provisionally accepted for publication in PLOS Computational Biology.

Best regards,

Ruy M. Ribeiro

Academic Editor

PLOS Computational Biology

Amber Smith

Section Editor

PLOS Computational Biology

Reviewer's Responses to Questions

**Comments to the Authors:**

Reviewer #1: The authors addressed all of my comments and questions very well.

Reviewer #2: Thank you for addressing the review suggestions so comprehensively.

**Have the authors made all data and (if applicable) computational code underlying the findings in their manuscript fully available?**

Reviewer #1: Yes

Reviewer #2: Yes

PLOS authors have the option to publish the peer review history of their article (what does this mean?). If published, this will include your full peer review and any attached files.

Reviewer #1: No

Reviewer #2: No

---

## [Editor Report · Acceptance letter]

PCOMPBIOL-D-24-01812R1

Clonal heterogeneity and antigenic stimulation shape persistence of the latent reservoir of HIV

Dear Dr Barton,

I am pleased to inform you that your manuscript has been formally accepted for publication in PLOS Computational Biology. Your manuscript is now with our production department and you will be notified of the publication date in due course.

With kind regards,

Zsofia Freund
